# Visible-broadband Localized Vector Vortex Beam Generator with a Multi-structure-composited Meta-surface

**DOI:** 10.3390/nano9020166

**Published:** 2019-01-29

**Authors:** Zhuo Yang, Dengfeng Kuang

**Affiliations:** Tianjin Key Laboratory of Optoelectronic Sensor and Sensing Network Technology, and Institute of Modern Optics, Nankai University, Tianjin 300350, China; nkyangzhuo@mail.nankai.edu.cn

**Keywords:** vortex beam, meta-surface, optical manipulation

## Abstract

We demonstrate a vortex beam generator meta-surface that consists of silver structures and graphene layers. The miniature material is just a few microns in size and the working part is only a few hundred nanometers thick. With the incidence of the linearly polarized beam, the meta-surface generates high-localized vector vortex beam with a high proportion of the longitudinal component. Being compared with the constituent part of the meta-surface, the multi-structure-combined meta-surface increases the localization by 250% and the longitudinal component proportion by 200%. Moreover, the above artificial material can generate vortex beams in broadband within the visible light range. These novel optical properties have the potential to improve the precision and sensitivity of nanoparticle manipulation. The study serves as a foundation in optical miniaturization and integration, nanoparticle manipulation, high-efficiency optical and quantum communication, and light-driven micro-tools.

## 1. Introduction

Vector vortex beams that carry orbital angular momentum [1,2,3,4] are widely utilized in particle manipulation and separation [5,6,7], optical communication [8,9,10,11], quantum computing [12,13] and even light-driven micro-tools [14]. Formerly, vortex beams were tuned by spiral phase plates [15,16], spatial light modulators [17,18] and computer-generated holograms [19,20]. Nowadays, meta-surfaces [21,22,23,24,25] are utilized to generate optical fields for the integration and miniaturization of optical components. For instance, Pancharatnam-Berry phase planar elements [26,27,28,29] and plasmonic vortex lens [30,31,32] realize the function of converting circularly polarized beams into vortex beams. In recent years, combinations of spirals and tapers were proposed for generation of localized vortex beams [6,33,34,35]. In addition, designed artificial structures were reported to convert linearly polarized beams into localized vector vortex beams [35,36]. In other words, it was good for nano-miniaturization and integration of optical elements that localized vector vortex beams can be generated without extra polarizers. Nevertheless, the proportion of the longitudinal component and the localization of transmitted fields generated by meta-surfaces above is not high enough in theory (<20%) [36], resulting in low-sensitivity and low-precision of particle manipulation, respectively. Besides, most vortex beam generators work only at several certain frequencies [28,29,35]. Therefore, broadband vortex beam generators are significantly useful in wider applications.

In this study, a vortex beam generator meta-surface is proposed. The multi-structure-composited meta-surface consists of silver structures and graphene layers. The miniature material is just a few microns in size and the working part is only a few hundred nanometers thick. With the incidence of the linearly polarized beam, the meta-surface generates high-localized vector vortex beam with a high proportion of the longitudinal component. Being compared with the constituent part of the meta-surface, the multi-structure-combined meta-surface increases the localization by 250% and the longitudinal component proportion by 200%. Moreover, the above artificial material can generate vortex beams in broadband within the visible light range. These novel optical properties have the potential to improve the precision and sensitivity of nanoparticle manipulation. This study serves as a foundation in optical miniaturization and integration, nanoparticle manipulation, high-efficiency optical and quantum communication, and light-driven micro-tools.

## 2. Materials and Methods 

The designed structure consists of a metallic dolphin-shaped cell circular array (MDCCA), metallic cylinder (MC) and graphene layer (GL). Simultaneously, ultrathin dielectric planar blocks are intended for substrates. In our study, materials of combinations are slightly different, including silver (in gray), graphene (in light purple) and silicon dioxide (in light blue) according to the legend that is inset of Figure 1. Dolphin-shaped cell meta-surface (DSCMS) is presented in Figure 1a. It consists of a MDCCA and dielectric substrate. Figure 1b,d,e show three types of combinations focused on in contemporary studies. Eight dolphin-shaped cells are angularly arranged to form MDCCAs, in the center of which the MCs are placed. The parameters R_1_ and R_2_ are defined as radii of the circular array and central metallic cylinder. The parameters d_1_, d_2_ and d_3_ are defined as the thickness of substrate, metal and graphene. The GLs, which consists of several layers of graphene monolayer, are added to the position in the interlayer of metallic structures and dielectric substrate. Figure 1g shows the geometric explanation of dolphin-shaped cell which is geometrically constituted by two inscribed cylinders of which radii are marked as r_1_ and r_2_ according to the Boolean operation. The dolphin-shaped cell can be defined as a surrounding area in two dimensions by two functions which are shown as follows:(1)$$f_{1}\left(x\right)=\{\begin{array}{ll} \sqrt{\frac{3}{4}r_{2}{}^{2}-r_{2}x-x^{2}} & ,x\in \left[-2r_{1}-\frac{r_{2}}{2},\frac{r_{2}}{2}\right)\\ -\sqrt{\left(2r_{1}+r_{2}\right)x-x^{2}-r_{1}r_{2}-\frac{1}{4}r_{2}{}^{2}} & ,x\in \left[\frac{r_{2}}{2},2r_{1}+\frac{r_{2}}{2}\right] \end{array}$$
(2)$$f_{2}\left(x\right)=\{\begin{array}{ll} \sqrt{-x^{2}-\left(2r_{1}+r_{2}\right)x-r_{1}r_{2}-\frac{1}{4}r_{2}{}^{2}} & ,x\in \left[-2r_{1}-\frac{r_{2}}{2},-\frac{r_{2}}{2}\right)\\ -\sqrt{\frac{3}{4}r_{2}{}^{2}+r_{2}x-x^{2}} & ,x\in \left[-\frac{r_{2}}{2},2r_{1}+\frac{r_{2}}{2}\right] \end{array}$$

The structural parameters of combinations are initialized as follows: *R*_1_ = 1000 nm, *R*_2_ = 500 nm, *r*_1_ = 150 nm, *r*_2_ = 300 nm, *d*_2_ = 100 nm, *d*_3_ = 35nm. Three-dimensional finite-difference time domain (3D-FDTD) methods are used in simulations. Perfectly matched layers are utilized as simulation boundaries with the incident linearly polarized beams of which the wavelength is 660 nm. Our own FDTD programs run with MATLAB 2013A (Nankai University, Tianjin, China). All of our simulations were performed on a PC machine (Nankai University, Tianjin, China) configured with an Intel 8-Core i7-7700 CPU @ 3.60 GHz (Nankai University, Tianjin, China).

## 3. Results and Discussions

Figure 2 shows the phase distribution of the longitudinal transmission fields which are generated by the four types of nano-structures shown in Figure 1a,b,d,e. It is obviously seen that all four nano-optical devices can generate vortex phases. However, there is a noteworthy difference in the radii of the vortexes at the same propagation distance. The incident beam passes through the central metal structure and the underlying graphene, causing a phase delay at the center, thereby causing a significant phase change at the edge of the central vortex field, which is significantly related to the localization of vortex beams. The characters above are illustrated in Figure 3. According to Formula A2, the phase delays are different as the beam passes through air and metal, causing a significant phase change shown in Figure 3. Figure 4a,c shows the amplitude of the optical field generated by a single metallic dolphin-shaped element to discuss the excitation process of the longitudinal field. The intensity of the incident light field is normalized. Surface plasmon polaritons (SPPs) were excited at the boundary of the metallic cells, which is explained by Formula A1 (Appendix A), then guided by the metallic element and finally squeezed to the tips to form highly localized strong electromagnetic fields of which the enhancement factor were greater than 2^15^. The strong fields radiated from the tips, then the vortex longitudinal field was generated.

Taking the simulation of the combination of MDCCA and MC as an example for characters of vortex beams, we kept the radius of circularly array fixed (*R*_1_ = 1000 nm) and varied the radius of MC from 0 to 500 nm with the step length of 100 nm. Figure 5a shows the radii of vortex phases (RVPs) generated by combinations of MDCCA and MC with different R_2_ and the same incident wavelength. For instance, at the distance *D* = 1000 nm the RVP stood at approximately 652 nm without the MC, 648 nm with *R*_2_ = 100 nm, 612 nm with *R*_2_ = 200 nm, 593 nm with *R*_2_ = 300 nm, 588 nm with *R*_2_ = 400 nm and only 518 nm with *R*_2_ = 500 nm. Therefore, the radius of MC plays a positive role in the localization of vortex phase generated by combination meta-surface. At its best, a MC roughly doubles the localization of the vector vortex beam. The localization of the vector vortex beam was significantly improved with *R*_2_ = 500 nm, but the localization decreased rapidly when the propagation distance was greater than 1800 nm. Similarly, Figure 5b shows the proportion of the longitudinal component of the transmitted field (PLT) generated by combinations of MDCCA and MC with different R_2_ and the same incident wavelength of 660 nm. As is shown in the line graph, among the six combinations with different R_2_, the PLT with *R*_2_ = 500 nm is the highest and that without the MC is the lowest at the distance below 2000 nm. This phenomenon is particularly evident when the propagation distance is between 1000 nm and 1600 nm. It can be seen that the radius of MC plays a positive role in the proportion of the longitudinal component of the transmitted field. At best, the PLT can be increased to about three times its value. When the particle is placed in the localized vortex field, the range of interaction between the particle and the light field decreased due to the reduction of the vortex area. Since the micro-optical device, such as the meta-surface, can precisely control the light field in sub-wavelength region and manipulate the particles placed in localized light field, the above-mentioned meta-surface with *R*_2_ = 500 nm exciting localized vector vortex beam can significantly improve the precision of particle manipulation. Moreover, the high proportion of the longitudinal component of transmitted fields can be obtained by the device above with *R*_2_ = 500 nm for ideal manipulation sensitivity because particle manipulation is closely related to the longitudinal component of vector optical fields. Figure 5c shows the RVPs utilizing different combinations with the same incident wavelength. At the distance *D* = 1400 nm, for instance, the RVP stood at approximately 690 nm with the combinations of MDCCA and GL, which was greater than that with the combination of MDCCA and MC (*R*_2_ = 500 nm). However, at the distance *D* = 2000 nm, the RVP stood at about 935 nm with the combination of MDCCA and GL, which was less than that with the combination of MDCCA and MC (*R*_2_ = 500 nm). This shows that the GL has a higher local potential for the vortex field at a further distance of propagation than the MC. In other words, using the GL as a part of the meta-surface above will result in greater particle manipulation accuracy at longer distances. At the distance *D* > 1300 nm, the RVPs with the combination of MDCCA, MC (*R*_2_ = 500 nm) and GL were less than that with the combination of MDCCA and one or two of the MC and GL, which means that the combinations of the three elements can produce highly localized vector vortex beams over a relatively long propagation distance. Figure 5d shows the PLTs when using different structures at different distances. The PLT through the GL was smaller than that without GL, which is related to the relatively high absorption of the graphene. At best, the GL increased localization by approximately 50%. Compared to individual devices, the combination of MDCCA, MC (*R*_2_ = 500 nm) and GL could increase localization by 250% and PLT by 200%. The binding of graphene layers to metals can improve localized resonances, which can improve the localization and the longitudinal component proportion simultaneously. When composite structures composed of three kinds of elements are used for particle manipulation, precision and sensitivity can be taken into account, which provides a foundation in optical elements design to build miniaturized and integrated optical systems of high-precision and high-sensitivity macromolecule manipulation.

We also study the vortex beam excitation characteristics of the composite devices in visible light region in order to expand the application in the wide-band wave-length range. Figure 6a shows the transmission spectrum of the combination of MDCCA and MC with *R*_2_ = 500 nm. On the whole, the transmittance was lower in the wavelength range of 600 nm and 650 nm than that in the other visible range. However, it will not affect the vortex beam generation because high-intensity micro-focus incident light can compensate for the loss caused by the low transmittance of the devices. In contrast, the PLTs excited by the devices in visible broadband directly determined whether the device can be applied for high-efficiency excitation of the localized vector vortex beam in visible light. Figure 6b shows the PLTs when using the combination of MDCCA and MC (*R*_2_ = 500 nm) in the incident wavelength range from 500 nm to 750 nm at different distances. The areas with a reddish color indicate that the PLTs were greater at the corresponding wavelength and propagation distance. The longitudinal component proportions were less than 70% at the region close to 550 nm. We hold the view that the proportion of light fields that are not regulated is larger because of the higher transmittance, causing the PVLs with smaller values. The device relies on transmitted excited fields to generate vortex beams. Unlike other common vortex beam generators, the device had very little limitation in the excitation wavelength band. It confirms that the devices can produce localized vector vortex beam in broad-band, which provides the guidance for the excitation and utilization of localized vector vortex light beams generated by the complex meta-surfaces above.

## 4. Conclusions

We demonstrate a vortex beam generator meta-surface that consists of silver structures and graphene layers. The advantages of the artificial material are as follows: The miniature material is just a few microns in size and the working part is only a few hundred nanometers thick. The side-length of the working part is less than 1500 nm. Moreover, the working part is approximately 135 nm thick.With the incidence of the linearly polarized beam, the meta-surface generates high-localized vector vortex beam with a high proportion of the longitudinal component. Being compared with the constituent part of the meta-surface, the multi-structure-combined meta-surface increases the localization by 250% and the longitudinal component proportion by 200%.The material can generate vortex beams in broadband within the visible light range. The results confirm that the devices can produce localized vector vortex beam in broadband, which provides the guidance for the excitation and utilization of localized vector vortex light beams generated by complex meta-surfaces above.Those above novel optical properties have the significantly potential to improve the precision and sensitivity of nanoparticle manipulation. The study serves as a foundation in optical miniaturization and integration, nanoparticle manipulation, high-efficiency optical and quantum communication, and light-driven micro-tools.

## Figures and Tables

**Figure 1 nanomaterials-09-00166-f001:**
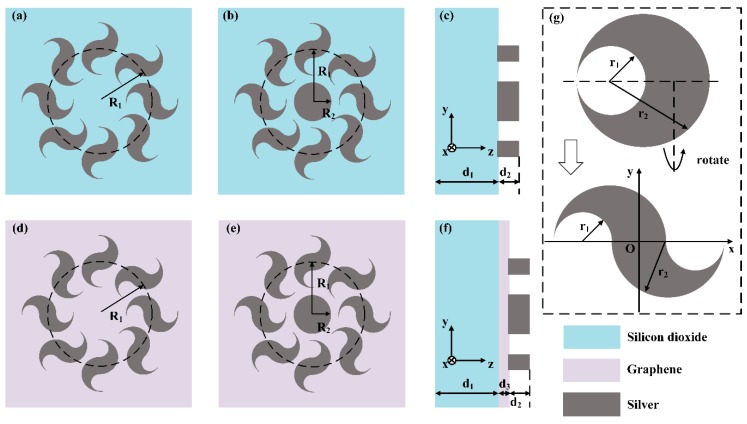
Schematic of the complex meta-surface which consists of metallic dolphin-shaped cell circular array (MDCCA), metallic cylinder (MC) and graphene layer (GL). (**a**) Schematic of dolphin-shaped cell meta-surface (DSCMS). (**b**) Schematic of combination of MDCCA and MC. (**c**) Lateral view of combination of MDCCA and MC. (**d**) Schematic of combination of MDCCA and GL. (**e**) Schematic of combination of MDCCA, MC and GL. (**f**) Lateral view of combination of MDCCA, MC and GL. (**g**) Geometric explanation of dolphin-shaped cell.

**Figure 2 nanomaterials-09-00166-f002:**
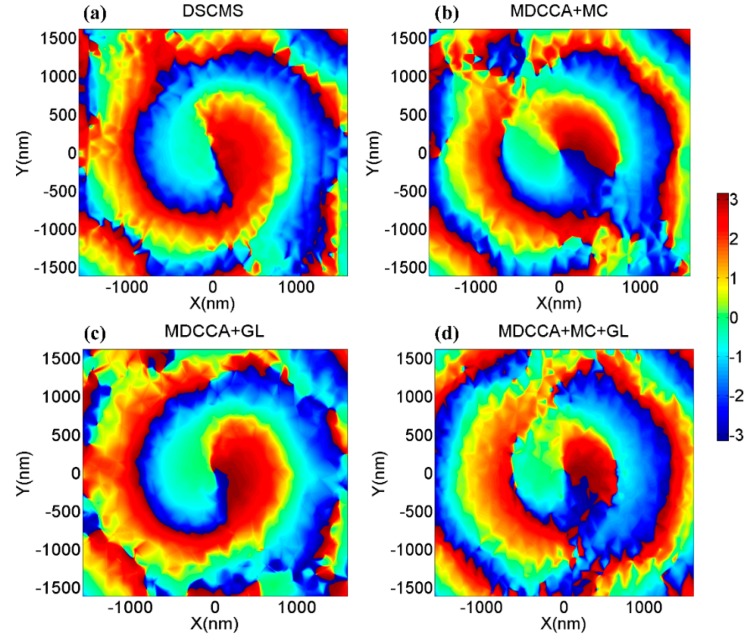
The phase distributions of the transmission fields of z components which are generated by (**a**) DSCMS, (**b**) combination of MDCCA and MC, (**c**) combination of MDCCA and GL, (**d**) combination of MDCCA, MC and GL at the distances *D* = 2000 nm.

**Figure 3 nanomaterials-09-00166-f003:**
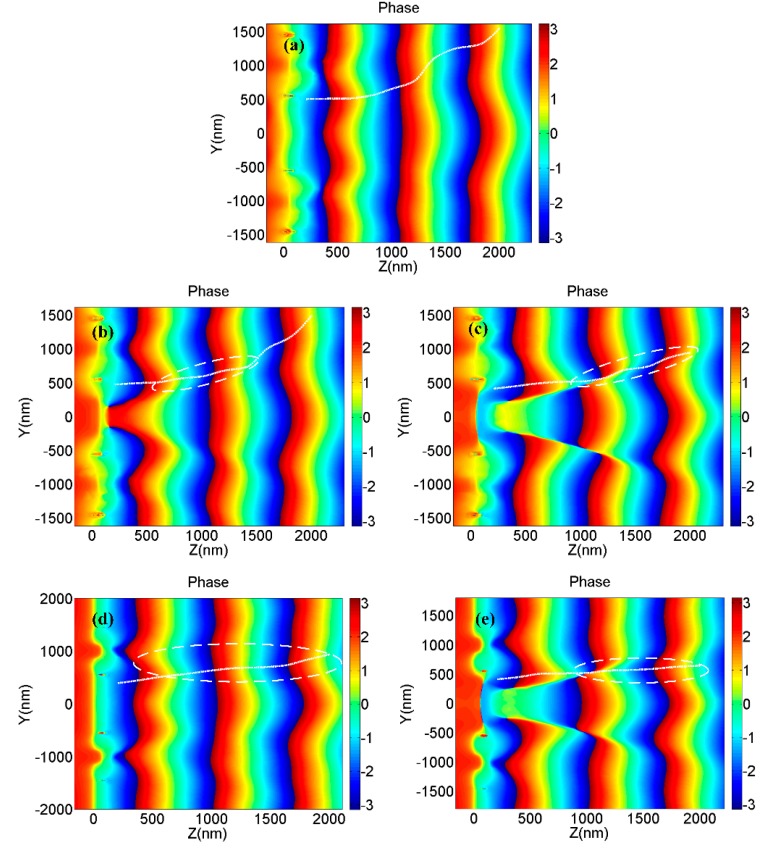
The phase distributions of the transmission fields generated by (**a**) DSCMS and combination of (**b**) MDCCA and MC (*R*_2_ = 300 nm), (**c**) MDCCA and MC (*R*_2_ = 500 nm), (**d**) MDCCA and GL, (**e**) MDCCA, MC (*R*_2_ = 500 nm) and GL at the plane *x* = 0, where the white solid lines are the edges of vortex phases and the white dashed lines indicate the coincident regions of vortex phase edges and phase mutations.

**Figure 4 nanomaterials-09-00166-f004:**
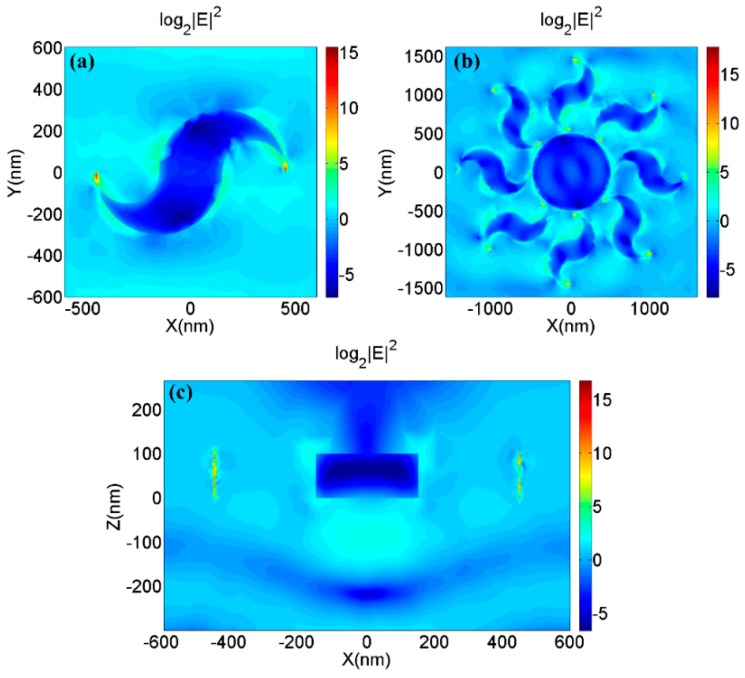
(**a**) The amplitude of optical field generated by a single metallic dolphin-shaped element at *z* = 100 nm and (**b**) that of optical field generated by the combination of MDCCA and MC. (**c**) The amplitude of optical field generated by a single metallic dolphin-shaped element at *y* = 0.

**Figure 5 nanomaterials-09-00166-f005:**
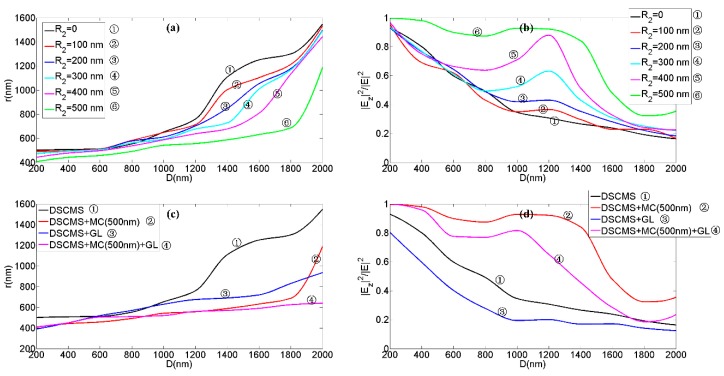
(**a**) The radii of transmitted vortex phases of the light fields generated by the combination of MDCCA and MC at different distances. (**b**) The proportion of the longitudinal component of the transmitted field at different distances when using the combination of MDCCA and MC. (**c**) The radii of transmitted vortex phases generated by different structures at different distances. (**d**) The proportion of the longitudinal component of the transmitted field when using different structures at different distances.

**Figure 6 nanomaterials-09-00166-f006:**
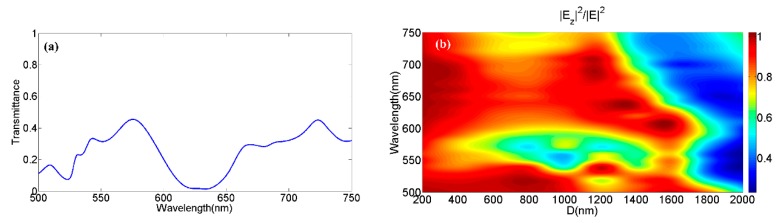
(**a**) Transmission spectrum of the combination of MDCCA and MC (*R*_2_ = 500 nm). (**b**) The PLTs when using the combination of MDCCA and MC (*R*_2_ = 500 nm) in the incident wavelength range from 500 nm to 750 nm at different distances.

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
