# Peer review of "Visible-broadband Localized Vector Vortex Beam Generator with a Multi-structure-composited Meta-surface"

_nanomaterials, 2019, doi:10.3390/nano9020166_

Reviewer 1 Report

Nice paper, almost ready for publication.

Some very minor comments:

I suggest to add some more words on the FDTD method you used. Did the authors use a commercial package? If they used their own, did they validate it? How?

Is there an effect of the direction of the incoming light?

I am not sure all the signs in eq. (2) are correct.

In figure 2, the authors comment on the different size of the vortexes, but I suggest to add a discussion on the fact that the phase distributions (b) and (d) appear cut.

Line 170: I would suggest to explain in more detail why the authors hold the view described here. 

A few words on the differences with reference [26] would be appreciated.

Author Response

Response to Reviewer 1 Comments

Point 1: I suggest to add some more words on the FDTD method you used. Did the authors use a commercial package? If they used their own, did they validate it? How? 

Response 1: Our own FDTD programs run with MATLAB 2013A of which the serial number was obtained by purchasing the legal software. All of our simulations were performed on a PC machine configured with an Intel 8-Core i7-7700 CPU @ 3.60 GHz. Relevant information has been added to the revised manuscript.

 Point 2: Is there an effect of the direction of the incoming light?

 Response 2: There’s no effect of the direction of the incoming light.

 Point 3: I am not sure all the signs in eq. (2) are correct.

 Response 3: Relevant information has been revised in the manuscript.

 Point 4: In figure 2, the authors comment on the different size of the vortexes, but I suggest to add a discussion on the fact that the phase distributions (b) and (d) appear cut.

 Response 4: According to Formula A2, the phase delays are different as the beam passes through air and metal, causing a significant phase change at the edge of the central vortex field. Relevant information has been added to the revised manuscript.

 Point 5: Line 170: I would suggest to explain in more detail why the authors hold the view described here.

 Response 5: The device relies on transmitted excited fields to generate vortex beams. Unlike other vortex beam generators, the device has very little limitation in excitation wavelength band. Relevant information has been added to the revised manuscript.

 Point 6: A few words on the differences with reference [26] would be appreciated.

 Response 6: The vortex beam generation in Ref. 24 relies on principles of Pancharatnam-Berry phase, which is different from our study. In our manuscript, surface plasmon polaritons (SPPs) are excited at the boundary of the metallic cells, then guided by the metallic element and finally squeezed to the tips to form highly localized strong electromagnetic fields of which the enhancement factor is greater than 215. The strong fields radiate from the tips, then the vortex longitudinal field is generated.

Reviewer 2 Report

At least in the version supplied to me I am still lacking a minus sign for the shape function f2 in the second line of equation (2). If this is fixed, I am fine with your edited manuscript.

Best regards

Author Response

Response to Reviewer 2 Comments

 Point 1: At least in the version supplied to me I am still lacking a minus sign for the shape function f2 in the second line of equation (2). If this is fixed, I am fine with your edited manuscript.

 Response 1: There’s something wrong about Formula 2 in our manuscript. Relevant information has been revised in our manuscript.

Round  2

Reviewer 1 Report

Thanks for taking my comments into account.

The manuscript is ready to be published. Good luck with your research!

This manuscript is a resubmission of an earlier submission. The following is a list of the peer review reports and author responses from that submission.

Round  1

Reviewer 1 Report

The submitted manuscript studies a nano-fabricated surface for designing the axial field of a light beam as an optical vortex beam.

Overall, I felt that the scientific content of this manuscript is very weak in several aspects:

1.       The description of the materials and methods is lacking fundamental information

a.       Are the authors showing results of experiments or simulations

b.      If simulations, what is the numerical method used; if experiments, what is the fabrication technique

c.       Nothing is said about the optical system used for device characterization: plane wave illumination? What about polarization? What about propagation distance in Fig. 2?

d.      We hardly understand that only the axial component of the electric field is characterized

e.      Nothing is said about the transverse component of the field.

f.        At Fig. 1, no scale bar is associated with drawings

2.       The proposed system is lacking any physical reasoning and argument

a.       No motivation is given for the geometry of the system (size, shape, number of dolphins...), for the materials used (silver, graphene, silicon dioxyde)

b.      No physical interpretation is given about results shown in Fig. 3-6, only qualitative comments of data.

3.       Nonsenses are suggested

a.       The authors discuss the optical vortex boundaries defined as a “phase jump”. It does not make sense. The amplitude may decay but a vortex does not have any boundary.

b.      The authors study the amplitude of the axial field with the propagation distance but the axial field does not propagate in the far field

Furthermore, the proposed design has already been published in a recent article (ref. 36).

In a nutshell, this manuscript could have been of interest but is lacking care and is thus inintelligible.

Reviewer 2 Report

 Dear authors, finding your article in principal interesting I have the following comments/suggestions:

1.      Did I get it wrong, or are all of your results based on simulations? If yes, I suggest pointing this out explicitly.

2.      Section 2) claims to treat "Materials and Methods" and I am missing a short description on the simulation methods used.

3.      Regarding the "dolphin" shaped geometry:

- I am lacking the size of the geometry in X- and Y-direction

- in the formula for the "shape function" f2, line 74, a minus sign is missing

- I suggest you indicate the X- or Y- and the Z-axis in one of the sketches (c) or (f) in Fig. 1, such that is becomes clear  from where you are measuring the distance "D" to which you are referencing in the explanation of e. g. Figs. 5 and 6.

4.      I would like to know for which polarization your simulations have been performed. A polarization in X- or Y-direction can be guessed, but please make this clear even though your suggested geometry should possess polarization independent properties due to discrete rotational symmetries.

5.      I cannot find any references to your appendix which raises the question whether it is needed.

Reviewer 3 Report

In this paper, the authors propose a nano-structure to generate localised vortex beams. They characterise this structure by presenting different optical parameters at different distances, in order to optimise the structure's performance.

During the review I found a good number of recent papers on the creation of vortex beams, and anyone who has used them in particle manipulation knows how important their beam parameters are. For this reason, the paper should be considered for publication. 

There is however a certain number of issues that need to be addressed, which appear quite obscure in the present version. My initial thoughts are listed below:

[General] The section on materials and methods does not contain any hint on how Figures 2-6 have been obtained. Are these simulations or measurements? How were the measurements taken?

[General] The design section does not explain how the authors came with the dolphin design.Figure 4a, which may help the reader in this sense, is barely commented in the text. It is not even clear why the authors decided to use 8 dolphins instead e.g. of a prime number.
Please modify accordingly.

[General] Please remember that readers may not print in colour. It might be worth adjusting pictures accordingly.

[Line 26] Not sure separateness is the best word here

[Line 29] Not sure about the use of "anticipant" here.

[Line 33] Please use "In addition" instead of "And then"

[Line 36] This text is crucial, as it states the limitations of the present methods, Would it be possible to quantify your claims? e.g. why is the ratio not high enough in theory? Who says that?

[Line 46] Broadband. The frequency response is a crucial aspect of any metamaterial structure. Can the authors state how wide in frequency do other systems work?

[Figure 2] How was this picture obtained? What did you measure? How? 
In addition, while it is clear that all the structure can create "a vortex beam", it is not at all clear which is the difference between 2a and 2c OR 2b and 2d. Please clarify.

[Figure 2 and 5] These two pictures how that while there is always a vortex beam, it is not always there at all distances. I strongly suggest the authors to take some additional measurements on another plane, to highlight the divergence of some beam conditions.

[Figure 3] This is not really commented in the text, so could be removed OR commented thoroughly. Also important is to clarify the reference axes, as they don't appear at all (and is therefore difficult to understand where "x=0" is.

[Line 101] how do you quantify localisation and efficiency? Please clarify

[Figure 5a] please clarify at which wavelength did you take the measurements? Also, use more contrasting pictures.

[Line 120] You did not prove any manipulation potential, so I would suggest to be more careful with claims.

[Line 131] Excellent claim!

[Line 138] "at best, the GL increases localization by XX" How is localisation defined?

[Line 158] Would it be possible to have a clear statement on how wide is the bandwidth

[Figure 5b] Do you have any explanation for the presence of a maximum as R2 is increased?

[Figure 6a] Tick Labels appear cut. 

[Figure 6b] Please comment on the implications of a "green region" close to 550 Hz. 
[Appendix A] I did not have a chance to review it...but I did not see it cited in the text either.